# The HIV Epidemic in South Africa: Key Findings from 2017 National Population-Based Survey

**DOI:** 10.3390/ijerph19138125

**Published:** 2022-07-01

**Authors:** Khangelani Zuma, Leickness Simbayi, Nompumelelo Zungu, Sizulu Moyo, Edmore Marinda, Sean Jooste, Alicia North, Patrick Nadol, Getahun Aynalem, Ehimario Igumbor, Cheryl Dietrich, Salome Sigida, Buyisile Chibi, Lehlogonolo Makola, Lwando Kondlo, Sarah Porter, Shandir Ramlagan

**Affiliations:** 1Human Sciences Research Council, Pretoria 0001, South Africa; mzungu@hsrc.ac.za (N.Z.); emarinda@hsrc.ac.za (E.M.); thili.sigida@gmail.com (S.S.); buyisile.chibi@caprisa.org (B.C.); lkondlo@hsrc.ac.za (L.K.); sramlagan@hsrc.ac.za (S.R.); 2School of Public Health, University of the Witwatersrand, Johannesburg 2000, South Africa; 3Human Sciences Research Council, Cape Town 8000, South Africa; lsimbayi@hsrc.ac.za (L.S.); smoyo@hsrc.ac.za (S.M.); sjooste@hsrc.ac.za (S.J.); anorth@hsrc.ac.za (A.N.); lmakola@hsrc.ac.za (L.M.); 4Department of Psychiatry & Mental Health, University of Cape Town, Cape Town 7925, South Africa; 5The School of Nursing and Public Health, College of Health Sciences, Howard College Campus, University of KwaZulu-Natal, Durban 4000, South Africa; 6School of Public Health and Family Medicine, University of Cape Town, Cape Town 7700, South Africa; 7Division of Global HIV and TB, U.S. Centers for Disease Control and Prevention, Pretoria 0001, South Africa; pcvnadol@gmail.com (P.N.); wpe5@cdc.gov (G.A.); wnz6@cdc.gov (E.I.); cheryledietrich@gmail.com (C.D.); icj9@cdc.gov (S.P.); 8School of Public Health, University of the Western Cape, Bellville, Cape Town 7535, South Africa

**Keywords:** population-based survey, HIV prevalence, HIV incidence, Antiretroviral (ARV) exposure, HIV testing, viral load suppression, risk behaviour, South Africa

## Abstract

South Africa has the largest number of people living with HIV worldwide. South Africa has implemented five population-based HIV prevalence surveys since 2002 aimed at understanding the dynamics and the trends of the epidemic. This paper presents key findings from the fifth HIV prevalence, incidence, and behaviour survey conducted in 2017 following policy, programme, and epidemic change since the prior survey was conducted in 2012. A cross-sectional population-based household survey collected behavioural and biomedical data on all members of the eligible households. A total of 39,132 respondents from 11,776 households were eligible to participate, of whom 93.6% agreed to be interviewed, and 61.1% provided blood specimens. The provided blood specimens were used to determine HIV status, HIV incidence, viral load, exposure to antiretroviral treatment, and HIV drug resistance. Overall HIV incidence among persons aged 2 years and above was 0.48% which translates to an estimated 231,000 new infections in 2017. HIV prevalence was 14.0% translating to 7.9 million people living with HIV. Antiretroviral (ARV) exposure was 62.3%, with the lowest exposure among those aged 15 to 24 years (39.9%) with 10% lower ARV coverage among males compared to females. Viral suppression among those on treatment was high (87.3%), whilst HIV population viral load suppression was much lower (62.3%). In terms of risk behaviours, 13.6% of youth reported having had an early sexual debut (first sex before the age of 15 years), with more males reporting having done so (19.5%) than females (7.6%). Age-disparate relationships, defined as having a sexual partner 5+ years different from oneself,) among adolescents were more common among females (35.8%) than males (1.5%). Self-reported multiple sexual partnerships (MSPs), defined as having more than one sexual partner in the previous 12 months, were more commonly reported by males (25.5%) than females (9.0%). Condom use at last sexual encounter was highest among males than females. Three quarters (75.2%) of people reported they had ever been tested for HIV, with more females (79.3%) having had done so than males (70.9%). Two-thirds of respondents (66.8%) self-reported having tested for HIV in the past 12 months. Finally, 61.6% of males in the survey self-reported as having been circumcised, with circumcision being more common among youth aged 15–24 years (70.2%), Black Africans (68.9%), and those living in both rural informal (tribal) areas (65%) and urban areas (61.9%). Slightly more (51.2%) male circumcisions were reported to have occurred in a medical setting than in traditional settings (44.8%), with more young males aged 15–24 (62.6%) and men aged 25–49 (51.5%) reporting to have done so compared to most men aged 50 and older (57.1%) who reported that they had undergone circumcision in a traditional setting. The results of this survey show that strides have been made in controlling the HIV epidemic, especially in the reduction of HIV incidence, HIV testing, and treatment. Although condom use at last sex act remains unchanged, there continue to be some challenges with the lack of significant behaviour change as people, especially youth, continue to engage in risky behaviour and delay treatment initiation. Therefore, there is a need to develop or scale up targeted intervention programmes to increase HIV testing further and put more people living with HIV on treatment as well as prevent risky behaviours that put young people at risk of HIV infection.

## 1. Introduction

The HIV epidemic in South Africa has been measured nationally using surveillance mechanisms and cross-sectional population-based HIV sero-prevalence surveys for more than 15 years [1,2,3,4,5]. These surveys, conducted every 3–5 years, help monitor the HIV epidemic, its impact, and its dynamics in the country [1,2,3,4,5,6,7]. Among the key findings from the fourth survey in 2012 was an increase in HIV prevalence to 12.2% compared to 10.6% in 2008 [7]. Five years later, in 2017, the fifth survey was conducted, continuing to provide valuable information on the epidemic.

The high prevalence of HIV in South Africa has been attributed to various biological, socio-behavioural, contextual, and structural drivers [3,4,8,9,10,11,12]. The majority of HIV transmission in South Africa is through heterosexual transmission [13], including through commercial sex. However, HIV prevalence is also increasing amongst men who have sex with other men [14]. Identified socio-behavioural and structural drivers of the HIV epidemic include having multiple sexual partners, condomless sex or inconsistent condom use, socio-economic power imbalances, harmful gender norms, sexual violence, alcohol and substance use [10,15,16]. Other socio-behavioural factors have been associated with HIV vulnerability-the age of sexual debut among youth and age-disparate relationships [17,18,19,20,21]. Age-disparate relationships between young women and older men have been driving the high prevalence of HIV in the country, especially amongst those aged 15–24 years [19,22]. This dynamic is evidenced by the eight-fold risk of HIV infection amongst adolescent girls and young women compared to their male counterparts [23].

New and strengthened interventions were implemented in 2012 to address these drivers of HIV transmission and to reduce morbidity and mortality amongst people living with HIV. Consistent and correct use of condoms has been found to reduce the risk of HIV infection. Initiatives such as the introduction of Max condoms ^®^ and the distribution of female condoms have been accelerated to improve accessibility [13]. There is overwhelming evidence showing that male medical circumcision reduces the risk of female-to-male HIV acquisition by 60% [24,25,26]. The implementation of the Voluntary Medical Male Circumcision (VMMC) programme was anticipated to reach 80% HIV non-infected males aged 15–49 years by 2015. However, VMMC uptake has remained lower than expected [13]. ARV has been successful in reducing deaths resulting from AIDS, reducing HIV transmission through viral suppression, as well as increasing life expectancy [27]. New guidelines on Universal Test and Treat (UTT) were released in South Africa in September of 2016 in order to expand the benefits of ARV to all people living with HIV.

In addition to an increase in HIV prevalence from 2008, some key findings from 2012 included information on the incidence of 1.07% in those 2+, prevalence, ARV exposure, HIV testing, and behavioural drivers. Antiretroviral (ARV) exposure was 31.2%, or two million people [7]. From 2005 to 2012, there was an increasing trend in uptake of HIV testing among men from 28% in 2005, 43% in 2008, and 59% in 2012 [28], albeit HIV prevalence also increased across those years in the same population. In 2012, risky behavioural factors such as engaging in multiple sexual partnerships were prevalent (12.6%) and fewer participants reported using a condom at their last sex compared to 2008 [7]. Updated information is required to accurately assess the impact of these efforts in recent years and understand the dynamics of the HIV epidemic in South Africa. In this paper, we present key findings of the fifth South African HIV household survey conducted in 2017–2018.

## 2. Materials and Methods

### 2.1. Design and Sampling

A complex multistage-stratified cluster sampling design was applied (for a detailed methodology, see [4,5,7]. The sample was stratified by province and geo-type where geo-type was categorised as urban, rural formal (farms), and rural informal (traditional tribal areas and rural villages) as described by Statistics South Africa [29]. A stratified master sample of 1000 small area layers (SAL) was randomly sampled with probability proportional to size, where the number of households, or visiting points (VPs), within SAL was used as a measure of size from the national sampling frame.

The 2017 survey design was similar to that implemented in the previous four surveys [1,2,3,4]. A systematic random sample of 15 households was sampled within each SAL. Detailed questionnaires for the household (administered to the head of the household) and age-appropriate individual behavioural questionnaires were administered to consenting/assenting eligible individuals and guardians answering on behalf of children under the age of 12 years. These questionnaires elicited information related to HIV knowledge, attitudes, practice, behaviours, and demographic information. Similar to both the 2008 and 2012 survey designs [3,4,7], persons of all ages living in the selected households and hostels were eligible to participate in the study. Respondents additionally consented to provide blood specimens to prepare dried blood spots (DBS). These DBS specimens were tested to determine HIV sero-status, HIV incidence, exposure to antiretroviral (ARV) drugs, viral suppression, and HIV drug resistance.

Data were digitally collected by interviewers from December 2016 to February 2018 using tablet computers with Census Survey Processing (CS Pro) software (US Census Bureau). Barcodes were used to anonymously link blood specimens to corresponding questionnaire data from respondents. Data were submitted to a central main-frame computer in the information technology (IT) department based at HSRC’s offices in Pretoria.

Two fourth-generation HIV-1 (enzyme immunoassays) EIAs, Roche Elecys HIV Ag/Ab assay (EIA 1) (Roche Diagnostics, Mannheim, Germany) and Genescreen Ultra HIV Ag/Ab assay (EIA 2) (Bio-Rad Laboratories, Hercules, CA, USA) were used to test for HIV antibodies. All HIV-positive samples were subjected to a nucleic acid amplification test (COBAS AmpliPrep/Cobas Taqman HIV-1 Qualitative Test, v2.0, Roche Molecular Systems, Branchburg, NJ, USA) to confirm HIV status. The HIV incidence algorithm used a Limiting-Antigen (LAg) Avidity EIA (Maxim Bio-medical, Rockville, MD, USA) in combination with information on ARV exposure and HIV viral load. Testing for exposure to antiretroviral drugs (ARVs) was performed by means of High-Performance Liquid Chromatography (HPLC) coupled with Tandem Mass Spectrometry for Nevirapine, Efavirenz, Lopinavir, Atazanavir, and Darunavir. Viral load measurements were conducted using the Abbott platform (Abbott m2000 HIV Real-Time System, Abbott Molecular Inc., Des Plaines, IL, USA).

The survey protocol was approved by the HSRC Research Ethics Committee (REC: 4/18/11/15) and the Division of Global HIV and TB (DGHT) of the U.S. Centers for Disease Control and Prevention (CDC).

### 2.2. Data Management and Statistical Analysis

SPSS was used for further data cleaning, exploration, and data management of survey data. Sampling weights were computed to account for unequal sampling at SAL level, household level, and individual response to the questionnaire and HIV testing to correct for potential bias due to unequal sampling probabilities. Final individual sampling weights were benchmarked to 2017 mid-year population estimates by age, race, sex, and province to provide population estimates [30] for analysis. Weighted data were analysed using Stata version 15 (Stata Corporation, College Station, TX, USA). Estimates of HIV prevalence, *p*-values, and confidence intervals (95% CI) are reported. Chi-squared analysis was used to test for associations and to assess trends in the HIV epidemic across the four surveys. HIV incidence was calculated as an annual instantaneous rate. The computational tools used were developed by the South African Centre for Epidemiological Modelling and Analysis (SACEMA) at Stellenbosch University [31].

## 3. Results

### 3.1. Response Rates

The 2017 survey targeted 15,000 visiting points (VPs), of which 12,435 (82.9%) were approached, and 11,776 (94.7%) VPs of these were found to be valid, resulting in a household response rate of 82.2% from the valid VPs. A total of 39,132 individuals from 9656 interviewed households were eligible to participate in the survey. Among the eligible individuals from these households, 93.6% agreed to be interviewed, with females slightly more (95.2%) likely to agree to be interviewed than males (92.3%). There were variations in interview participation by race, such that Black Africans 24,701 (95.6%) had the highest response rate, followed by Coloureds 6646 (92.3%), Whites 2292 (89.7%), and Indians/Asians 2767 (82.1%). Of those who were eligible to be interviewed, 61.1% further provided blood specimens for biomedical tests. Of the 61.1% of participants who agreed to provide a blood specimen, females were more likely (64.3%) to provide a blood specimen when compared to males (57.7%). By age group, the highest testing response rate was among those aged 15 to 24 years (66.9%), followed by 50 years and older (64.4%), 25 to 49 years (61.5%), 2 to 14 years (57.3%) and least among those under 2 years of age (48.1%). The HIV testing response rate was highest in rural formal (farms) areas at 65.1% and lowest in urban areas (57.3%).

### 3.2. HIV Incidence

HIV incidence in 2017 was estimated at 0.48% per year (95% CI: 0.42–0.54%) among people aged 2 years and older, translating to 231,100 [95% CI: 211,900–260,400] new infections in 2017. The incidence of HIV among females was 0.51% (95% CI: 0.43–0.59%) and 0.46% (95% CI: 0.39–0.51%) among males, translating to 121,900 [95%CI: 102,800–141,000] and 109,200 [95%CI: 92,600–121,100] newly infected individuals respectively. Figure 1 shows the incidence of HIV by age and sex. Thirty-eight percent of new infections were found among youth aged 15 to 24 years. Females in this age group had the highest burden at 1.51% (95% CI: 1.31–1.71%) and accounted for the majority of new infections among youth.

The incidence of HIV among those that were married was 0.61% (95% CI: 0.51–0.71%); among those who were single, it was 1.07% (95% CI: 0.95–1.19%) and was 0.96% (0.86–1.06%) among those that were going steady/living together with their partners. Black African females aged 20–34 years had the highest incidence of HIV at 1.59% (95% CI: 1.41–1.77%). Incidence among Black African males aged 25–49 years was 0.97% (95% CI: 0.85–1.09%).

### 3.3. HIV Prevalence

The national estimate of HIV prevalence in South Africans of all ages was 14.0% (95% CI: 13.1–15.0), translating to an estimated 7.92 (95% CI: 7.1–8.8) million people living with HIV. Figure 2 shows the epidemiological profile of HIV prevalence by age and sex in 2012 and 2017. Among females, HIV prevalence continues to increase among those aged 30 years and above, indicating an ageing pattern of the epidemic with no noticeable change among females younger than 30 years. A similar pattern is observed among males older than 35 years, albeit a noticeable decline in HIV prevalence among those aged between 25 and 35 years.

### 3.4. Exposure to Antiretroviral (ARV) and HIV Viral Load Suppression

In 2017, out of over 7.9 million people living with HIV, 4,402,000 (62.3%) were exposed to ART. A greater proportion of females (65.5%; 95% CI: 62.4–68.4%) than males (56.3%; 95% CI: 51.0–61.5%) living with HIV were on treatment. Half of the HIV-positive children aged 0–14 years (50.0%; 95% CI: 36.6–63.3%) were found to have been exposed to ARV. Exposure to ARV was highest among those aged 50 years and above (76.7%; 95% CI: 71.3–81.4%), intermediate for 25–49-year-olds (63.1%; 95% CI: 59.2–66.8%), and lowest among youth (39.9%; 95% CI: 32.1–48.3%). Among the 7.6 million Black Africans estimated to be living with HIV, 62.6% (95% CI: 59.5–65.6%) were found to be on treatment. The proportion of ARV exposure in other race groups was 51.3% (95% CI: 36.4–66.1%). Geographically the KwaZulu-Natal Province had the highest number of people on ARV.

Among the 2994 people whose specimens tested HIV positive, 2946 (98.4%) had a viral load result available. Of those with a viral load result, 62.3% (95% CI: 59.5–65.0%) were virally suppressed (<1000 copies HIV RNA/mL). This implies that 37.7% of people living with HIV were not virally suppressed and thus more likely to transmit HIV during unprotected sexual intercourse. Viral load suppression was higher among females (66.5%; 95% CI: 63.5–69.3%) compared to males (55.0%; 95% CI: 50.1–59.9%). Viral load suppression varied by age at 51.9% (95% CI: 41.1–62.5%) among those 0–14 years, 47.7% (95% CI: 39.5–56.1%) among 15–24 years, 62.8% (95% CI: 59.3–66.1%) among 25–49 years and 73.2% (95% CI: 67.2–78.5%) among those aged 50 years and older.

Viral load suppression among people with ARV, as expected, was high (87.3%) compared to 62.3% in the general population that is irrespective of ARV exposure. Viral load suppression among those on ARV was consistently above 85% in most subgroups (e.g., locality, provincial, and age) of people living with HIV. The lowest viral suppression levels among those on ARV were among males (82.4%), 0–14-year-olds (81.9%), and people living with HIV living in formal rural areas (82.6%).

### 3.5. UNAIDS 90-90-90 Targets

An estimated 84.9% (95% CI: 81.7–87.7%) of people living with HIV aged 15 to 64 years old knew their HIV status (1st 90) at the time of the survey. Amongst those aware of their HIV status, 70.6% (95% CI: 67.4–73.6%) were on ARV (2nd 90), and among those on ARV, 87.5% (95% CI: 84.9–89.7%) had suppressed viral load (3rd 90). This yields an achievement of 85-71-88 for the UNAIDS 90-90-90 targets. Females were more likely to be aware of their HIV status than males (88.9% [95% CI: 85.9–91.4%] versus 78.0% [95% CI: 72.1–82.9%]). The proportion of males and females who were aware of their status and on ARVs did not differ. Seventy-two percent of females who were aware of their status (95% CI: 68.8–75.3%) were on ARV, as were 67.4% of their male peers (95% CI: 61.9–72.4%). However, women on ARV had better viral suppression levels (89.9% [95% CI: 87.1–92.2%]) than their male counterparts (82.1% [95% CI: 76.5–86.6%]).

### 3.6. Related Behavioural Determinants

We present results on the following behavioural drivers of HIV: early sexual debut, age-disparate relationships, multiple sexual partners, and lack of condom use. Concerning early sexual debut (first sex before the age of 15 years) among both males and females (Reported for respondents aged 15–24 years), 13.6% (95% CI: 12.1–15.3%) reported having had early sexual debut, with more males reporting that they had done so more (19.5%; 95% CI: 17.0–22.3%) than females (7.6%; 95% CI: 6.1–9.4%). As for age-disparate relationships (Reported for respondents aged 15–19 years), defined as having a sexual partner 5 or more years older than oneself, among adolescents aged 15–19 years, it was found to be more common among females (35.8%; 95% CI: 29.7–42.3) than males (1.5%; 95% CI: 0.6–3.6%). Self-reported multiple sexual partnerships (MSPs) (Reported for respondents aged 15+ years), defined as having more than one sexual partner in the previous 12 months, were more commonly reported by males (15.4; 95% CI: 13.9–17.0) compared to females (5.0%; 95% CI: 4.5–6.3%). Finally, the data show that condom use at the last sexual encounter was highest among people aged 15–24 years and declined in subsequent age groups. When examining marital status by age group, the highest prevalence of condom use at last sex was 68.2% (95% CI 64.7–71.6%) for those aged 15–24 years. The lowest was 11.2% (95% CI: 8.5–14.8%) of married respondents aged 50 and older who used a condom during the last sex. Furthermore, reported condom use was significantly higher among males than females aged 15–24 years (67.7% vs. 49.8% respectively) and those aged 25–49 years (40.2% vs. 36.0%).

### 3.7. HIV Testing

In total, 75.2% (95% CI: 74.0–76.4) had been ever tested for HIV, with more females (79.3%; 95% CI: 78.0–80.5) having done so than males (70.9%; 95% CI: 69.2–72.5). Slightly more urban respondents (77.4%; 95% CI: 76.0–78.7%) had ever been tested for HIV compared to respondents living in rural informal (70.3%; 95% CI: 68.0–72.4%) and rural formal areas (71.2; 95% CI: 67.2–74.9%). Two-thirds of respondents (66.8%; 95% CI: 65.4–68.2%) self-reported being tested for HIV in the past 12 months. The proportions of males and females self-reporting a recent HIV test were similar.

### 3.8. Male Circumcision

Overall, 61.6% (95% CI: 59.3–63.9%) of males in the survey reported having been circumcised. The highest rate of circumcision was reported by those aged 15–24 years (70.2%), followed by those aged 25–49 years (61.8%) and those aged 50 and older (54.0%). The circumcision rate was highest among Black Africans (68.9%; 95% CI: 66.4–71.2%) compared to the other three race groups (Whites, Coloured, and Indians/Asians range 26.3–32.5%). Circumcision was also higher in both rural informal (tribal) areas (65%) and urban areas (61.9%) than in formal rural areas (45.5%). A slight majority (51.2%; 95% CI: 48.0–54.4%) of male circumcisions were reported to have occurred in a medical setting than in traditional settings (44.8%; 95% CI: 41.5–48.1%). More circumcised males aged 15–24 years (62.6%) and 25–49 years (51.5%) reported circumcision in medical settings to most men aged 50 years and older (57.1%) who reported that they had undergone circumcision in a traditional setting.

## 4. Discussion

South Africa has made strides in the fight against the scourge of the HIV epidemic, as shown in the key findings from the fifth South African survey of HIV prevalence, incidence, and behaviour, conducted in 2017. However, South Africa has remained unchanged and has regressed in other areas, thus delaying the possibility of realising the end of the HIV epidemic in the immediate future.

The findings from this survey show that compared to previous years, the incidence of HIV has declined from 1.07% in 2012 to 0.48% per year in 2017 [7]. All age-sex groups reflect this decline. However, the HIV incidence remains high enough to sustain the HIV epidemic. These findings are consistent with those of the Thembisa model [32]. Although annual incidence in young women has declined significantly since 2012, they continue to carry the burden of HIV incidence. Young women aged 15 to 24 years contributed 66,200 of all new HIV infections in 2017. Recent sub-national population-level estimates of HIV incidence using a combination of synthetic cohort and recency biomarker approaches in KwaZulu-Natal, South Africa, also found young females continue to be newly infected with HIV at high rates [33]. The high burden could be due to age-disparate relationships [22,34], as well as the burden young females face in accessing testing and treatment services and condom use [35].

HIV prevalence has increased over the years in South Africa, especially among adults. This increase should be interpreted in the context of expanded access to ARVs which can lead to reduced HIV-related mortality coupled with sustained new HIV infections [7]. The shift in the epidemiological curve indicates that those who are infected with HIV continue to live longer. Other findings confirm this reduction of mortality among people living with HIV, with an estimated 10.5% yearly decrease in HIV deaths in South Africa from 2007–2017 [36]. Unfortunately, large numbers of people are still getting infected with HIV, especially among youth, where the prevalence of HIV has remained unchanged since 2008 [7]. This unchanging pattern in those younger than 30 years indicates a lack of success in HIV prevention.

Access to ARV is increasing among people living with HIV. However, there are gender and age disparities in accessing ARV. Females are more likely to seek and access treatment than males, whilst older people are more likely to be on treatment than those that are young, a finding that remains unchanged from 2012 [37]. One contributing factor to this dynamic is that males are less likely to seek medical attention when needed, therefore delaying HIV diagnosis and treatment initiation [38]. Females were more likely to have tested for HIV compared to males in this study. The differentials in treatment by age raise equity concerns that are critical to the fulfilment of the sustainable development goal three [37,39]. The delayed diagnoses and treatment among youth are likely to lead to an increased risk of morbidity and increased onward transmission of the virus. Less than two-thirds of people living with HIV were on ARV in months following the policy to put all those diagnosed with HIV on treatment. Continued monitoring of ARV initiation and adherence will be needed to assess the impact following full implementation of the Treat All campaign that allows for immediate ARV among those that are infected with HIV. Additionally, failure to initiate and adhere to ARV has been attributed to individual, social and structural barriers. These include time needed to accept HIV status, religious beliefs, HIV stigma, disclosure issues, especially to family and partners, access, negative attitudes towards healthcare workers, and communication challenges between healthcare workers and patients [40,41,42,43]. Interventions to address these barriers may be required to achieve access and adherence to treatment for all people living with HIV, the onward transmission of the virus, and mortality.

Access and adherence to ARV increase the likelihood of viral load suppression. Viral load suppression among those on ARV was 87.3% compared to 62.3% of everyone living with HIV (irrespective of access to ARV). This translates to 37.7% of people living with HIV not being virally suppressed in the general population highlighting the scale of treatment coverage gap that can contribute to sub-optimal outcomes at the individual level as well as on-going transmission of HIV. Viral suppression increases with age due to older people being more likely to be on ARV compared to those that were young. Among youth not on ARV, 77.5% were virally unsuppressed, indicating a high risk of HIV transmission among youth during unprotected sexual intercourse. Youth aged 15 to 24 need to be specifically targeted to be enrolled in treatment with strengthened adherence programs that will lead to higher levels of viral suppression as a way to prevent further transmission of HIV [44].

A number of sexual behaviours contribute to an increased risk of HIV infection. These include early sexual debut, age-disparate relationships, multiple sexual partners, and lack of condom use [45]. The results show an increase in the proportion of youth reporting early sexual debut. Women were more likely to report having had a sexual partner 5 years older than themselves, which has been shown to increase the risk of HIV among youth [19]. Males were more likely to report multiple sexual partners in the last 12 months. Furthermore, condom use remains a challenge as low levels of condom use at last sex continue to be reported. The role of confounding factors in the use of condoms, such as marital status and inter-partner relations, need to be considered in the interventions aimed at improving condom use [46,47]. For behaviours related to biomedical prevention, self-reported circumcision was highest (70.2%) among males aged 15 to 24 years. For the first time since this survey collected circumcision information, male medical circumcision surpassed traditional male circumcision, although overall VMMC uptake is lower than the 80% target.

The results presented from this large, population-based survey are designed to be applied to the respective populations of South Africa. The overall HIV testing response rate was lower in 2017 compared to the 2012 survey (67.5%). Although, this was still within an acceptable range to draw generalizable inference [48], and efforts are underway to conduct a more rigorous analysis of the impact of such differential uptake. The declining HIV testing response rate does present some limitations for inferences among Whites and Indians.

An analysis of the impact of such bias on the 2012 survey indicated no significant difference in HIV prevalence based on such differences in testing uptake by age and sex [49].

## 5. Conclusions

There remain high rates of HIV incidence, especially among youth, that continue to sustain the HIV epidemic in South Africa. Access to ARV differs considerably by age and gender, thus translating to age and gender differentials in the levels of viral suppression. There is a need to expand access to treatment among youth. Evidence-based HIV prevention interventions are urgently required to help reduce the incidence and thus the prevalence of HIV among those younger than 30 years, with a particular focus on youth.

## Figures and Tables

**Figure 1 ijerph-19-08125-f001:**
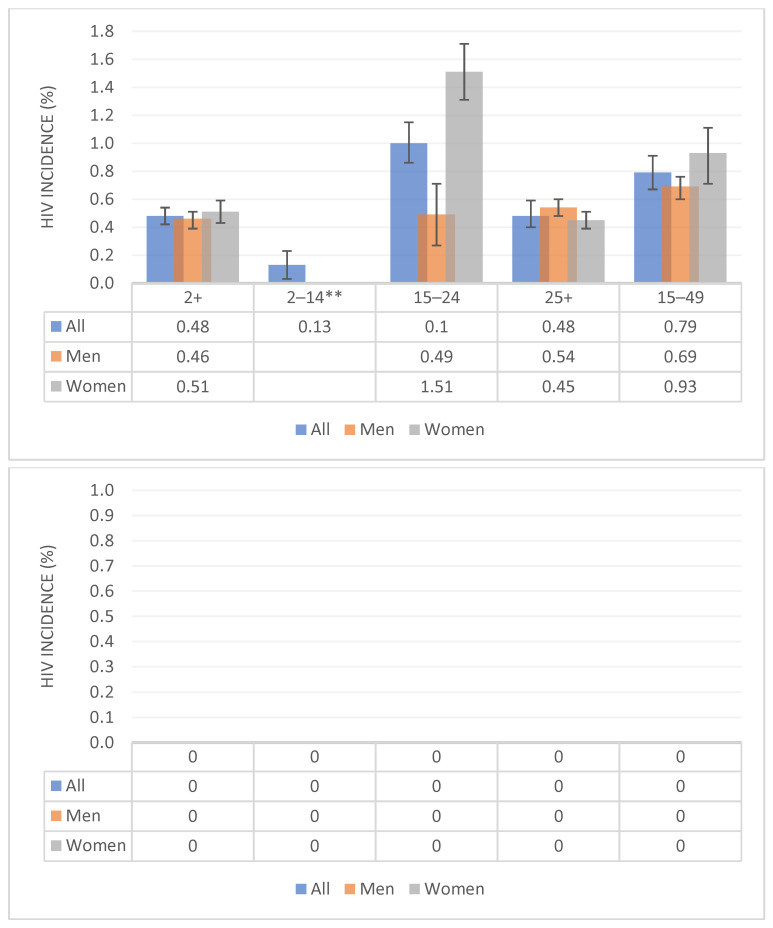
Incidence of HIV by age and sex, South Africa 2017. ** The sample for 12–14 was too small to stratify by sex.

**Figure 2 ijerph-19-08125-f002:**
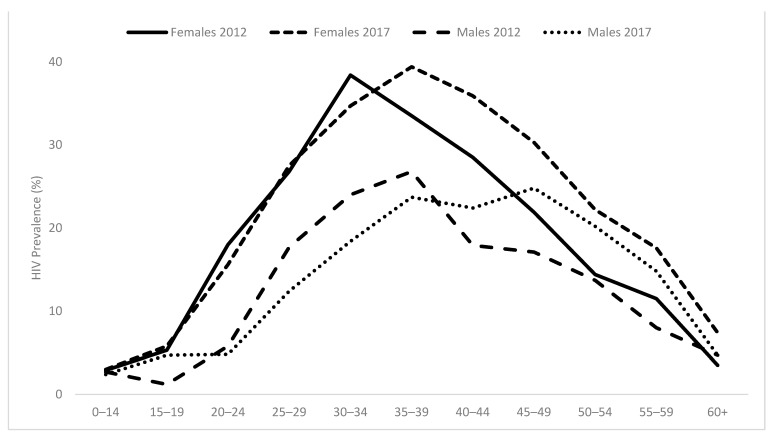
HIV prevalence by age for males and females in South Africa, 2012 and 2017.

## Data Availability

The data for this manuscript are openly available on the Human Sciences Research Council institutional repository available at https://repository.hsrc.ac.za/handle/20.500.11910/15468 (accessed on 16 May 2022), Archive number: SABSSM 2017 Combined, doi:10.14749/1585345902.

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
