# Peer review of "The HIV Epidemic in South Africa: Key Findings from 2017 National Population-Based Survey"

_ijerph, 2022, doi:10.3390/ijerph19138125_

Round 1

Reviewer 1 Report

Thank you for the opportunity to review your study. The paper is well written and organized, and I learned a lots about HIV situation in South Africa.

Abstract and Introduction: Did ARV exposure include those who took PrEP?

Materials and Methods: dates of data collection. 

Results:

-Table 1 with characteristics of survey participants (e.g., race, city/state, age, gender) would help readers understand the intended population. Response rates were reported, but we would also like to know how many African Americans, Coloureds, Whited, and Indians/Asians were in your samples.

-Line 198: why did you strict to aged 2 years?

-Survey respondents included people with HIV, right? You reported the HIV incidence in 2017 was estimated at 0.48%. The proportion did not include those who knew their HIV seropositive status prior to the survey correct? Please add clarification in the Materials and Methods section.

-Line 228, did you also survey over 7.9 million PLWH or surveillance data? If the numbers were estimated based on your survey with math calculation, please describe them.

-Line 239, were 2,994 people whose specimens tested HIV+ not aware of their HIV seropositive status? Or did you test for HIV all no matter of their HIV status. Please clarify how many 2,994 people knew their HIV status so they started ARV treatment.

-Figure 1: is there incidence rate for Men and Women in aged 12-14? The first category is 2+ or 2-11?

Discussion:

-Line 324, is there any reasons you can think of why young women aged 15 to 24 years had such a burden of HIV incidence?

Reviewer 2 Report

The authors report HIV epidemiologic findings from the fifth national based survey conducted in 2017 in South Africa. Overall HIV incidence was 0.48% in 2017 which was lower than the result from fourth survey conducted in 2012. However, HIV prevalence has increased over years to 14% in 2017, translating to 7.9 million people living with HIV. Access to ARV was higher among females than males, older people than young people. This manuscript compares the findings from 2017 with those from 2012, and overall it reinforces the importance of ARV treatment and the need to reduce HIV incidence nationwide.

Major:

It would be great if the authors and make a statistic table listing all the key findings mentioned in the manuscript. The readers are easier to follow.

Minor:

1. Several extra spaces are found in the manuscript (e.g. lines 76, 106, 135, 250, 372, etc.), please delete them. 

2: The subtitle in line 227 should be “Exposure to ARV exposure and HIV viral load suppression”. There’s an extra “exposure”.

3. References 6 and 7 are missing.
